# Brain Evaluation by Dual PET/CT with [^18^F] FDOPA and [^18^F] FDG in Differential Diagnosis of Parkinsonian Syndromes

**DOI:** 10.3390/brainsci14090930

**Published:** 2024-09-18

**Authors:** Fabio Andrés Sinisterra Solís, Francisco Rubén Romero Castellanos, Emilly Alejandra Cortés Mancera, Ana L. Calderón Ávila, Sofía Denisse González Rueda, Juan Salvador Rosales García, Nora Estela Kerik Rotenberg, Dioselina Panamá Tristán Samaniego, Andrés Mauricio Bonilla Navarrete

**Affiliations:** 1PET/CT Molecular Imaging Unit, National Institute of Neurology and Neurosurgery, Mexico City 14269, Mexico; franciscoromerocastell@gmail.com (F.R.R.C.); juansrosales_esmipn@hotmail.com (J.S.R.G.); nkerikfmmn@gmail.com (N.E.K.R.); 2Nuclear Medicine Department, National Cancer Institute, Mexico City 14080, Mexico; denissegr205@gmail.com; 3PET/CT Molecular Imaging Unit, Salud Digna, Mexico City 04100, Mexico; 4Nuclear Medicine Department, Hospital de Especialidades del Centro Médico Nacional Siglo XXI, IMSS, Mexico City 06720, Mexico; 19.anacalderon@gmail.com; 5Adult Neurologist and Movement Disorders Specialist, Ministry of Health, Panama City 512-9100, Panama; dradioselina@gmail.com; 6Department of Clinical Neurology, National Cancer Institute of Colombia, Bogotá 111511, Colombia; andresm.bonilla@urosario.edu.co

**Keywords:** dual PET/CT, FDOPA, FDG, Parkinsonian syndrome, differential diagnosis

## Abstract

Parkinsonian syndromes are considered clinicopathological conditions that are challenging to diagnose. Molecular imaging with [18F]-FDOPA and [18F]-FDG contributes to a more accurate clinical diagnosis by evaluating presynaptic dopaminergic pathways and glucose metabolism, respectively. The aim of this study was to correlate diagnoses made from dual PET/CT with the initial clinical diagnoses, as well as during follow-ups in patients with Parkinsonian syndromes. A secondary objective was to describe the imaging findings. **Methods:** A total of 150 patients with a clinical diagnosis of neurodegenerative Parkinsonism were evaluated using dual PET/CT. Clinically, 82% were diagnosed with PD, while the remaining 18% had an atypical Parkinsonism. **Results:** Using dual PET/CT, the most frequent diagnosis was PD in 67% of the patients, with the rest being diagnosed with an atypical Parkinsonism. In an agreement analysis between the initial clinical diagnosis and the imaging diagnosis by dual PET/CT, a concordance of 94.1% (n = 95) was observed for PD. In the remaining patients, the clinical diagnosis differed from that suggested by dual PET/CT, with atypical Parkinsonian syndromes being diagnosed as DLB in 40% (n = 4), PSP in 46.7% (n = 7), MSA-C in 75% (n = 6), MSA-P in 70% (n = 7), and CBD in 66.7% (n = 4). A total of 38.66% (n = 58) of patients were followed up (median follow-up of 27 months), with a Kappa coefficient of 0.591 (*p* < 0.001), suggesting substantial agreement. **Conclusions:** Dual FDOPA–FDG PET/CT demonstrated moderate agreement with the initial clinical diagnosis of Parkinsonism and moderate to substantial agreement during follow-up. This dual technique, therefore, stands out in differentiating between types of Parkinsonisms.

## 1. Introduction

Neurodegenerative Parkinsonian syndromes, including Parkinson’s disease (PD) and atypical Parkinsonisms, such as Progressive Supranuclear Palsy (PSP), Multiple System Atrophy (MSA), Dementia with Lewy Bodies (DLB), and Corticobasal Degeneration (CBD), form part of a broader spectrum of movement disorders. Although these conditions are distinct clinicopathological entities, they share overlapping clinical and neuropathological features, which complicates the diagnostic process despite the availability of well-established criteria [1,2]. Neuropathological examination remains the gold standard for diagnosis; however, this is not feasible in clinical practice since a definitive diagnosis can only be confirmed postmortem [3]. Molecular imaging with biomarkers such as [^18^F]-L-Dihydroxyphenylalanine (FDOPA) and [^18^F]-Fluorodeoxyglucose (FDG), which allow for in vivo assessment of the presynaptic dopaminergic pathway and brain glucose metabolism, respectively, play crucial roles in the differential diagnosis of these disorders [4,5].

These radiotracers, however, represent just a fraction of the available molecular imaging options. Numerous radiolabeled molecules have been developed to evaluate various aspects of the dopaminergic pathways, including those targeting aromatic amino acid decarboxylase, vesicular monoamine transporters, dopamine transporters, and postsynaptic dopamine receptors (D1 and D2) [6,7]. While several studies have documented the diagnostic accuracy of these radiopharmaceuticals in evaluating neurodegenerative Parkinsonisms, their ability to provide a definitive diagnosis on their own is limited. Moreover, only a few studies have explored the impact of dual-tracer protocols [8,9]. Some have used these protocols to differentiate PD from MSA [10], while others have employed dual-phase protocols with lipophilic tracers, which allow for an early perfusion phase that simulates brain glucose metabolism, followed by a late phase representing the molecular target binding phase [11]. Recent advancements in positron emission tomography (PET) have led to the development of tau-specific radiotracers, offering new opportunities in the differential diagnosis of atypical Parkinsonisms, particularly in conditions such as PSP and CBD. First-generation tau tracers like [18F]-AV-1451 have shown variable specificity for 4R tauopathies, characteristic of these disorders. However, it is important to note that [18F]-AV-1451 and similar first-generation tracers also present off-target binding properties, particularly their affinity for monoamine oxidase (MAO), which can reduce imaging specificity. Second-generation tracers, including [18F]-PI-2620 and [18F]-PM-PBB3, have demonstrated improved specificity and efficacy in detecting tau pathology in PSP and CBD, overcoming some of the limitations observed with earlier tracers. Although promising, further research with larger cohorts is needed to fully establish the diagnostic utility of these tracers in clinical practice [12].

The aim of this study is to correlate dual FDOPA–FDG PET/CT diagnoses with the initial clinical diagnosis, as well as to assess its consistency during follow-up in patients with Parkinsonian syndromes. A secondary objective is to describe the imaging findings.

## 2. Materials and Methods

### 2.1. General Description

This study was conducted following approval and authorization by the Research Ethics and Biosafety Committee under protocol number 97/22 at the National Institute of Neurology and Neurosurgery “Dr. Manuel Velasco Suárez”. We retrospectively reviewed medical records and brain dual-tracer PET/CT (FDOPA–FDG) images obtained between January 2018 and December 2021. Sociodemographic data and clinical diagnoses were collected based on initial assessments by neurologists specialized in movement disorders, with follow-up data recorded up until 30 November 2022. Clinical diagnoses were established by a panel of expert neurologists during medical board meetings, using updated diagnostic criteria for each Parkinsonian syndrome: the International Parkinson and Movement Disorder Society (MDS) Clinical Diagnostic Criteria for PD [13], the MDS Clinical Diagnostic Criteria for PSP [14], the MDS Clinical Diagnostic Criteria for MSA [15], the Fourth Consensus Report of the DLB Consortium for the Diagnosis and Management of DLB [16], and the Armstrong–Litvan Criteria for CBD [17].

Patients aged 18 years and older were included if they met the following criteria: a clinical diagnosis of neurodegenerative Parkinsonism, including established or probable PD, probable or possible PSP, probable or possible DLB, probable or possible CBD, and probable or possible MSA in its Parkinsonian (MSA-P) and cerebellar (MSA-C) variants. Eligible patients also had to have complete medical records and have undergone PET/CT scans with FDG and FDOPA within 12 months of each other. Exclusion criteria included patients diagnosed with non-neurodegenerative Parkinsonisms (neoplastic, vascular, toxic, pharmacologic, or other etiologies), those with PET/CT scans performed more than 12 months apart, or those with suboptimal image quality that precluded adequate medical interpretation. Analyzed variables included the dual brain PET/CT imaging diagnosis, age, gender, clinical diagnosis prior to the PET scans, findings on FDOPA PET, findings on FDG PET, the level of concordance between the dual PET/CT diagnosis and the clinical diagnosis, and the association between the maintenance of the clinical diagnosis during follow-up and the dual PET/CT diagnosis.

### 2.2. Imaging Protocol

PET/CT brain imaging acquisitions were performed in a hybrid Siemens Biograph 64 mCT (Siemens Medical Solutions). Dual-tracer FDOPA and FDG were intravenously injected at fixed activities of 185–370 MBq (5–10 mCi) for both radiopharmaceuticals, blood glycemia of ≤150 mg/dl, with at least 4 h of fasting. For the FDOPA scan, 150 mg of carbidopa was administered orally 60 min prior to the radiotracer administration. Dynamic and sequential images were acquired over 90 min. FDG preparation consisted of minimal stimuli, faint lightning, 30–45 min for biodistribution, and wide-open eyes.

The computed tomography (CT) component was performed for attenuation correction and anatomic location, monophasic, 80–100 kV, and 350–380 mA. Transverse PET sections were reconstructed through attenuation correction, based on CT, using an iterative algorithm (ordered subset expectation maximization [OSEM] + time of flight [TOF] + point spread function [PSF]) and a full width at half maximum of 5 mm and 5 iterations.

### 2.3. Imaging Analysis

FDOPA–FDG dual PET/CT images were evaluated by two nuclear medicine physicians with extensive experience in nuclear neurosciences. They reached a consensus to establish the diagnosis based on imaging patterns, conducting a simultaneous and qualitative visual analysis. This approach was adopted to maximize diagnostic accuracy by integrating the complementary information provided by both radiotracers, thereby overcoming the limitations inherent to each radiotracer when used alone. It is important to note that the nuclear medicine physicians were blinded to the clinical diagnosis to avoid bias, although they were aware that the images came from patients with movement disorders.

For each of the neurodegenerative Parkinsonisms, FDOPA and FDG imaging findings were meticulously analyzed, relying on well-established patterns in the scientific literature.

**FDOPA PET:** Uptake patterns were assessed based on symmetry and asymmetry in the striatal regions, particularly in the putamen and caudate nuclei. The putamen was subdivided into anterior and posterior regions to evaluate presynaptic dopaminergic dysfunction more precisely, as described in the literature for PD and other atypical Parkinsonisms [18]. Decreased tracer uptake in the striatal nuclei suggests an alteration in the presynaptic dopaminergic pathway, indicative of a neurodegenerative Parkinsonism. Conversely, normal uptake makes a diagnosis of neurodegenerative Parkinsonism less likely [19,20].

**FDG PET:** Cerebral metabolism was assessed using the same structural configuration of the striatum as in FDOPA PET and included an examination of specific cortical regions. These regions were divided into hemispheres, lobes, and specific surfaces: dorsolateral frontal cortex, medial frontal cortex, dorsolateral temporal cortex, mesial temporal structures, dorsolateral parietal cortex, dorsolateral occipital cortex, anterior and posterior cingulate, precuneus, cuneus, and cerebellar hemispheres. These metabolic patterns are crucial for differentiating between various neurodegenerative Parkinsonisms, as documented in the literature [21,22].

Although the nuclear medicine physicians used statistical mapping tools as additional support for analyzing the findings, final diagnosis was primarily based on their clinical expertise and qualitative visual analysis of the images.

### 2.4. Statistical Analysis

For data analysis, IBM SPSS Statics (24th version) was utilized. For descriptive statistics of quantitative variables, median and percentiles were used according to a non-normal distribution, using Kolmogorov–Smirnov’s test. For qualitative variables, data are presented according to frequencies and percentages. For inferential analysis and comparison between groups, Pearson’s x^2^ test, Fisher’s exact test, and chi-square test for linear trend were used. For the agreement analysis between imaging diagnosis by dual PET/CT, initial clinical diagnosis and diagnosis through follow-up, the Kappa index was used. Statistical significance was established with a *p*-value of <0.05.

## 3. Results

### 3.1. General Characterization of the Studied Population

A total of 480 patients underwent dual PET/CT imaging in the context of movement disorders. After applying the inclusion and exclusion criteria, 150 patients were selected for this study. The median age at diagnosis among these patients was 64 years old, with a slightly higher prevalence in males (54.7%) compared to females. Clinically, 82% of the patients were diagnosed with PD, while the remaining 18% were diagnosed with atypical Parkinsonisms, with PSP and DLB being the most prevalent among them at 7.4% and 4.7%, respectively (Table 1).

### 3.2. Brain [^18^F]DOPA PET/CT Findings

Among the 150 patients, 149 (99.3%) exhibited abnormal findings on FDOPA PET/CT. The most common abnormality observed was a reduction in uptake in the posterior third of the putamen nuclei, which occurred in 89.9% of cases. Only one patient had normal FDOPA PET results, with both clinical diagnosis and dual PET/CT suggesting DLB. An asymmetrical uptake pattern was the most common, seen in 65.1% (n = 97) of cases. Among the different Parkinsonisms, MSA-C was the only diagnosis that displayed both symmetrical and asymmetrical patterns. The asymmetrical pattern was most common in MSA-P (Figure 1).

### 3.3. Brain [^18^F] FDG PET/CT Findings

Disturbances in glucose metabolism were observed in 47.33% of the patients. Normal metabolic findings were more frequent in patients diagnosed with PD, where overall normal cortical metabolism was observed alongside normal or hypermetabolism in the basal ganglia and hypometabolism in the parietal, occipital dorsolateral, and orbitofrontal cortex. In patients with atypical Parkinsonisms, metabolic alterations primarily affected the basal ganglia, with almost all cases presenting hypometabolism, except for DLB, where hypermetabolism in the striatum nuclei was also documented (Figure 2).

### 3.4. Dual FDOPA–FDG PET/CT Diagnosis

The most frequent diagnosis provided by dual PET/CT was PD, identified in 67% (n = 101) of the patients. The remaining 33% were diagnosed with atypical Parkinsonisms: PSP in 10% (n = 15), DLB and MSA-P in 7% (n = 10) each, MSA-C in 5.33% (n = 8), and CBD in 4% (n = 6). The findings from FDOPA and FDG PET scans are summarized in Figure 1.

### 3.5. Agreement Analysis between Initial Clinical Diagnosis and Dual PET/CT Diagnosis

The concordance analysis between the initial clinical diagnosis and the diagnosis derived from dual PET/CT showed a high level of agreement, with a Kappa coefficient of 0.447 (*p* < 0.0001). Specifically, 94.1% (n = 95) of PD diagnoses were consistent between clinical and dual PET/CT evaluations. However, dual PET/CT identified atypical Parkinsonisms in some patients initially diagnosed with PD: four cases were reclassified as DLB, seven as PSP, six as MSA-C, seven as MSA-P, and four as CBD. Conversely, dual PET/CT reclassified six patients with an initial atypical Parkinsonism diagnosis to PD (Table 2).

During follow-up (median 27 months), 38.66% (n = 58) of the patients were reassessed. In this stratified analysis, there was a 93.5% (n = 29) agreement between the initial clinical diagnosis and the dual PET/CT diagnosis in patients with PD during follow-up (*p* < 0.001). Two patients continued to be diagnosed with PD despite dual PET/CT suggesting DLB. Among patients with an initial diagnosis of DLB, 66.7% (n = 4) maintained their diagnosis in subsequent evaluations (*p* = 0.102), while two patients were reclassified as PSP. In PSP cases, 60% (n = 3) maintained their initial diagnosis across all evaluations, although two cases were reclassified as PD and MSA-P during follow-up (*p* = 0.350). All patients with MSA-C (n = 2) and CBD (n = 1) maintained their initial diagnosis in subsequent evaluations (Table 3).

## 4. Discussion

The concordance between the initial clinical diagnosis and the dual PET/CT diagnosis is essential in the approach to assess movement disorders, where clinical manifestations may overlap and discriminating between different entities can be challenging. In this study, we observed an overall concordance between both diagnostic approaches of 94.1% in the diagnosis of PD. The high level of concordance found emphasizes the utility of dual PET/CT as a reliable diagnostic tool in the identification of PD.

Nevertheless, it is essential to recognize the disparities that emerged in our results. In a significant number of patients initially diagnosed with PD, dual PET/CT identified atypical Parkinsonisms, such as DLB, PSP, MSA, and CBD. These discrepancies underscore the importance of a comprehensive evaluation and the use of next-generation imaging to achieve precise differentiation of movement disorders, particularly in atypical cases.

We also performed a stratified analysis between dual PET/CT diagnosis and clinical diagnosis at follow-up to assess the evolution of the clinical entity. We found that in the majority of cases with a diagnosis of PD by dual PET/CT, there was a high level of concordance with both clinical diagnoses (initial and follow-up). These findings advocate for the consistency of the PD diagnosis over time in this set of patients, supporting the accuracy of dual PET/CT in identifying this disorder.

However, we also observed some differences during follow-up, where some patients experienced a change in diagnosis despite a previously established initial clinical diagnosis. For example, some patients initially diagnosed with PD were later reclassified as having DLB or MSA-P because of dual PET/CT, highlighting the need for continuous evaluation and reevaluation of patients with movement disorders over time.

Research on the utility and relevance of dual PET/CT with FDOPA and FDG in the differential diagnosis of neurodegenerative Parkinsonisms is very limited, highlighting the uniqueness and high value of our study in this field. To our knowledge, the only article that has addressed this topic similarly is by Emsen et al. [9]; their findings show a high frequency of abnormalities on FDOPA PET, as well as the ability of dual PET/CT to provide additional diagnostic information in a significant number of cases, which supports and complements our results.

In 2021, the study conducted by Xian et al. [10] investigated the analysis of a coregistered set of dual PET/CT images with FDOPA and FDG in the differential diagnosis between PD and MSA-P, compared to healthy individuals, demonstrating the crucial value of the dual method in this aspect, which is perhaps the clinical scenario that represents the greatest diagnostic challenge. In our cohort, dual PET/CT changed the clinical diagnosis from PD to MSA-C in six patients and to MSA-P in seven patients; it also changed the clinical diagnosis from MSA-P to PD. This information offers exceptional value for better classifying these patients, with the goal of establishing a more accurate prognosis and, consequently, providing better treatment and follow-up. (Figure 2, Figure 3 and Figure 4).

Some authors, such as Minyoung et al. [11], have explored the utility of dual-phase PET/CT with [^18^F]-FP-CIT, a radiopharmaceutical that, because of its lipophilic characteristics, allows the acquisition of an initial perfusion phase, which is compared with cerebral glucose metabolism, followed by a late phase that directly evaluates the presynaptic dopaminergic pathway. They assessed the diagnostic performance of dual-phase PET/CT with both radiopharmaceuticals ([^18^F]-FP-CIT and [^18^F]-FDG) compared to the clinical diagnosis in a cohort of 141 patients, including subjects with PD, MSA, PSP, and individuals without movement disorders as controls; their results showed similar performance between the dual-phase with FP-CIT (95.7%) and the dual-phase, dual-tracer technique (FDG in the early phase and FP-CIT in the late phase, 97.2%). However, they observed decreased performance when evaluating FP-CIT and FDG separately. These findings suggest that both acquisition phases in dual-phase PET with FP-CIT can provide valuable diagnostic information comparable to that obtained by dual-tracer PET/CT (FDOPA and FDG).

When analyzing the findings of PET/CT per radiopharmaceutical, we found that, regarding FDOPA, 149 out of 150 patients presented some type of alteration in the presynaptic dopaminergic pathway, with a greater defect in the posterior third of the putamen (89% of cases); the most frequent pattern was asymmetrical in 64.7%. The only Parkinsonism that exhibited a symmetric uptake defect pattern was MSA-C; on the other hand, only one patient with a clinical diagnosis of DLB showed integrity in the presynaptic dopaminergic pathway, although the uptake pattern on FDG PET was consistent with that of the clinical diagnosis. In 2020, Stormezand and collaborators reported data that supported the findings of decreased uptake on FDOPA PET in both idiopathic PD and atypical Parkinsonisms, with predominance in the posterior third of the putamen [20].

Nurmi and Brousolle reported in different studies that PD first shows alterations in the posterior putamen, followed by the anterior putamen and caudate nuclei, involving the side contralateral to the clinically affected one [23,24]. In our cohort, all PD patients presented a presynaptic dopaminergic pathway alteration; according to the literature, up to 10–20% of clinically diagnosed PD cases have shown integrity of the presynaptic dopaminergic pathway when evaluated by PET [25]. This situation can also occur with DLB, in which a study demonstrates that integrity of the presynaptic dopaminergic pathway does not rule out the diagnosis [26]; this is similar to the example mentioned above, where the diagnosis was established by the altered pattern in cerebral glucose metabolism, highlighting the importance of dual studies when clinical suspicious is high.

It has been described that in MSA-P, the pattern of decreased uptake with the FDOPA tracer is primarily evident in the caudate nuclei and anterior putamen [27]. This information differs from our findings, where, as mentioned earlier, the posterior putamen was the most affected. In the case of CBD, the alteration of the striatal nuclei on the side opposite to the clinically affected hemibody has been described [28], similar to our findings. Normal findings have also been reported; however, in our study, we did not have any normal findings in this pathology.

Regarding FDG PET/CT findings, 52.67% of patients showed preserved metabolism, which can be explained by the fact that most patients had a clinical and dual PET/CT diagnosis of PD, in whom cerebral glucose metabolism is generally preserved. Metabolic patterns with FDG have been useful in the differential diagnosis of PD, as well as in atypical Parkinsonisms [29,30]. These findings correspond with those described by Garraux et al., who concluded that FDG PET/CT accurately distinguishes between PD and atypical Parkinsonisms, a relevant factor for prognosis and treatment [31]. Figure 1 summarizes the findings from dual FDOPA and FDG PET/CT in our population; until now, this had not been characterized in Mexico, and the results were similar to published data.

In our experience, the evaluation of movement disorder cases using dual FDOPA–FDG PET/CT provides higher diagnostic accuracy than when each tracer is evaluated separately. FDOPA imaging, when abnormal, indicates dysfunction in the presynaptic dopaminergic pathway, suggesting the presence of a neurodegenerative Parkinsonism and ruling out non-neurodegenerative causes. However, FDOPA alone cannot differentiate between various neurodegenerative Parkinsonisms. On the other hand, FDG, by evaluating cerebral glucose metabolism, can differentiate between atypical Parkinsonisms based on specific metabolic patterns, but only in the context of a confirmed neurodegenerative Parkinsonism, which requires prior confirmation with FDOPA (or any other radiotracer that evaluates the dopaminergic presynaptic pathway).

When analyzing the PET/CT findings in this study, it was observed that the PD diagnosis by dual PET/CT had a high level of concordance with the initial and follow-up clinical diagnoses. These findings support the accuracy of dual PET/CT in identifying PD and suggest that this dual imaging approach can serve as a reliable tool for consistent diagnosis over time. However, there were cases in which the initial clinical diagnosis of PD was later changed to an atypical Parkinsonism after dual PET/CT, a diagnosis that was subsequently confirmed by clinical follow-up. This highlights the importance of continuous evaluation of patients with movement disorders and the value of early diagnosis through dual PET.

Our study demonstrates the importance of dual FDOPA–FDG PET/CT in the differential diagnosis of neurodegenerative Parkinsonisms, particularly in distinguishing between PD and atypical Parkinsonisms. This dual imaging approach is critical for achieving early and accurate diagnoses, which are essential for optimizing patient management and treatment outcomes.

We acknowledge that some limitations emerged in our study, such as the retrospective nature of the study and the relatively moderate sample size of atypical Parkinsonisms cases, which are known to be less common than PD. However, it should be noted that the initial population was 480 patients, and only 150 patients met all the inclusion criteria. Additionally, while postmortem verification remains the gold standard for diagnosis, it is impractical in routine clinical settings, and we used the initial and follow-up clinical diagnoses using established and updated clinical criteria for each of the movement disorders included here. On the other hand, qualitative visual analysis provides a powerful tool for evaluating PET/CT images, but it is limited to medical training and experience in this area, as well as inter-observer variability, so further research can benefit from the incorporation of automated tools or artificial intelligence algorithms to complement visual analysis, thereby improving diagnostic accuracy and reducing inter-observer variability.

Parkinsonian syndromes are complex entities that require comprehensive approaches, and molecular imaging offers a wide range of radiotracers that allow for in vivo assessment of different levels of the presynaptic and postsynaptic dopaminergic pathway, cerebral glucose metabolism, and cerebral perfusion. These allow for the combination of different evaluation strategies, such as combining the assessment of the presynaptic pathway with perfusion or cerebral metabolism, combining the presynaptic and postsynaptic dopaminergic pathways, etc. Our study was based on complementing the assessment of the dopaminergic pathway with cerebral glucose metabolism. Some authors have conducted studies comparing these different strategies, showing comparable results. Future research should include the comparison of these strategies with new radiotracers that are being evaluated, targeting specific neuropathological alterations such as tau and α-synuclein.

## 5. Conclusions

Dual FDOPA–FDG PET/CT exhibited a moderate to substantial agreement with the initial clinical diagnosis of neurodegenerative Parkinsonisms, as well as the diagnosis during follow-up; our findings stand out the relevance of dual PET/CT as a valuable diagnostic tool in the distinction of Parkinsonisms, highlights the reduction in the number of patients diagnosed with PD, and, simultaneously, augments the number of patients diagnosed with atypical Parkinsonisms. This capability of dual PET/CT to distinguish among diverse subtypes of atypical Parkinsonisms and PD is the bottom line to the optimal management of these patients, highlighting the urgent relevance of its inclusion in the comprehensive diagnostic evaluation of these neurological disorders. At last, as a graphical summary, we propose the following clinical and imaging diagnostic algorithm to aid in the differential diagnosis of the different types of Parkinsonisms using dual FDOPA–FDG PET/CT.



## 6. Patents

This section is not mandatory but may be added if there are patents resulting from the work reported in this manuscript.

## Data Availability

Data are contained within the article.

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
