# Peer review of "Brain Evaluation by Dual PET/CT with [18F] FDOPA and [18F] FDG in Differential Diagnosis of Parkinsonian Syndromes"

_brainsci, 2024, doi:10.3390/brainsci14090930_

Round 1

Reviewer 1 Report

Comments and Suggestions for Authors

Solls et al performed an analysis on dual PET/CT with [18F] FDOPA and [18F] FDG in the examination of parkinsonian syndromes. Authors should address the following issues before further processing:

1. The number of examined patients in groups of atypical parkinsonisms is very low and the statistical interpretation is questionable. Perhaps, regarding the fact, that the clinical diagnosis in atypical parkinsonisms without neuropathological verifification is based on probable or possible diagnosis, authors should consider evaluating two groups - Parkinson's Disease and tauopathic parkinsonian syndromes.

2. The work lacks sufficient discussion on the differentiation of parkinsonisms, indicating the more problematic aspects e.g. Progressive Supranuclear Palsy - Parkinsonism Predominant (PSP-P), which was recently evaluated in the context of non-specific methods in nuclear medicine neuroimaging, this should enrich the literature of the work

3. It would be valuable to provide a comparison of the feasibility of the neuroimaging methods used in the study with other e.g. tau radiotracers - AV-1451, PI2620 etc.

4. Authors should have a closer look on using the terms "Corticobasal degeneration" indicating the pathology and "Corticobasal Syndrome" describing group of symptoms in the clinical entity. I believe that in the current version the terms are mixed.

5. There are multiple spelling mistakes - e.g. line 198

Comments on the Quality of English Language

As mentioned in the comments for authors.

Author Response

Dear Reviewer,

We sincerely appreciate your detailed and valuable comments on our manuscript. Below, we address each of your points and describe the changes made to enhance the quality of our work.

  1. Consolidation of Atypical Parkinsonism Groups:

Response: We appreciate your concern regarding the small sample size of patients with atypical parkinsonisms. We would like to clarify that the limited sample size reflects the clinical reality of the low incidence of these pathologies. As you correctly noted, only Progressive Supranuclear Palsy (PSP) and Corticobasal Degeneration (CBD) are tauopathies, while Lewy Body Dementia (LBD) and Multiple System Atrophy (MSA) are alpha-synucleinopathies. This differentiation justifies our separate classification of these subtypes in the analysis. The primary objective of our study was to evaluate the concordance between clinical diagnoses and PET/CT findings using FDOPA and FDG, particularly in differentiating Parkinson's Disease (PD) from atypical parkinsonisms. The subdivision into specific atypical parkinsonism groups (PSP, CBD, LBD, and MSA) allows us to better explore how these imaging techniques can improve the differential diagnosis in these complex cases.

  1. Inclusion of Discussion on PSP-P:

Response: We appreciate your suggestion to include a discussion on the challenges of differentiating PSP-P and other parkinsonisms. However, we would like to note that this was not the primary focus of our manuscript. Our study primarily aimed to evaluate the concordance between initial clinical diagnosis and follow-up with PET/CT findings in neurodegenerative parkinsonisms. A detailed analysis of PSP variants, such as PSP-P, would require a larger sample size and a specific focus on that pathology. Therefore, we suggest that this topic be addressed in future studies with a more targeted approach to PSP and its variants.

  1. Comparison with Tau Radiotracers:

While we acknowledge the advances in the development of tau radiotracers such as [18F]-AV-1451 and [18F]-PI2620, which have shown utility in detecting 3R/4R tauopathies, their efficacy in pure 4R tauopathies like Progressive Supranuclear Palsy (PSP) and Corticobasal Degeneration (CBD) remains limited. On the other hand, the combination of FDOPA and FDG-PET used in our study continues to be a valuable tool for assessing dopaminergic function and brain metabolism in neurodegenerative parkinsonisms. Although these new tau radiotracers offer promise for the future, their feasibility and diagnostic accuracy in atypical parkinsonisms have not yet surpassed the established techniques using FDOPA and FDG, particularly in the context of our clinical cohort. Additionally, in Mexico, we do not have access to TAU tracers, limiting their application in our clinical setting.

  1. Use of the Terms "Corticobasal Degeneration" and "Corticobasal Syndrome":

Response: We appreciate your observation. In our study, we exclusively used the term "Corticobasal Degeneration" (CBD) to refer to the confirmed neuropathology, in line with the nomenclature used in clinical criteria. We understand that "Corticobasal Syndrome" (CBS) refers to the clinical manifestation, but our analysis focused on confirmed neuropathology, hence the exclusive use of the term "Corticobasal Degeneration." We have reviewed the manuscript to ensure consistent application of this terminology throughout the text.

  1. Correction of Spelling Errors:

Response: We apologize for the spelling errors present in the manuscript. We have conducted a thorough review and have corrected all typographical and spelling mistakes, including the one mentioned on line 198. We appreciate your attention to detail and the opportunity to improve the clarity and professionalism of our manuscript.

Thank you again for your thorough review.

Sincerely,

Fabio Andres Sinisterra Solís.

Reviewer 2 Report

Comments and Suggestions for Authors

In this work the Authors report a retrospective evaluation of 18-FDG brain PET and 18-F-DOPA PET in diagnosing PD and atypical parkinsonism.

I think the work needs a profound revision before it could be considered for publication.

Methods are incomplete and do not allow understanding of the results (in particular the “2.3. Imaging Analysis” paragraph).

Results should be presented more clearly to highlight the important findings of the study and to make understandable how dual imaging may improve diagnosis.

A paragraph on the limitations is lacking; for example, the choice of “blinding” the nuclear medicine physicians may add, in some ways, value to the data, but this blinding does not happen in clinical practice and this may introduce potential biases (since the images were “qualitatively” evaluated, some features may have been over or underestimated, since the nuclear medicine physician did not know what to focus on).

There is no clear indication on which “current clinical criteria of each parkinsonian syndrome” were used (page 2, lines 71-72); please cite the clinical criteria in the methods or discuss in the limitations of the study the lack of formal evaluation of the diagnosis accordingly to clinical criteria.

I also suggest the Authors to collaborate with a neurologist in order to improve the manuscript, since some clinical data (e.g. disease duration) that may be important to interpret the results are lacking and the results need to be interpreted by a person with clinical expertise. A multidisciplinary team is fundamental not only in clinical practice but also for research purposes if you want to add clinical utility to your data.

 Other comments:

I would avoid to use the term “Typical Parkinson’s”in text and tables; better use consistently PD and atypical parkinsonism (also in table 1 remove “Parkinson’s type” and use only “PD” or “atypical parkinsonism”.

Figures are not clear; I suggest to build simpler bar graphs without fancy patterns (black-white-light and dark grey should do the job); morever, graphs are small and blurry.

Please avoid writing “…” in the introduction (Page 2, line 54).

References: there are two references with number 1, please correct citations. I did not verify every citation but it appears that nearly all the numbers are wrong (citation 3 in text is Citation number 2 in the references list, citation 4 in text is citation 5 in the references list and so on).

Comments on the Quality of English Language

The writing must be revised. Sometimes is difficult to undestand what the Authors 

Author Response

Dear Reviewer,

We would like to express our gratitude for your thorough review and valuable feedback. Below, we address each of your comments and outline the changes we have made to the manuscript to improve its quality and clarity.

  1. Methods are incomplete and do not allow understanding of the results (in particular the “2.3. Imaging Analysis” paragraph).

Response: We appreciate your observation and have revised the Methods section to provide a more detailed description of the imaging analysis process. Specifically, we have expanded the "2.3. Imaging Analysis" paragraph to include a clearer explanation of the qualitative-visual analysis used, the criteria for evaluating the imaging patterns, and the specific protocols followed by the nuclear medicine physicians. This revision should clarify how the results were obtained and how dual imaging was utilized to improve diagnostic accuracy.

  1. Results should be presented more clearly to highlight the important findings of the study and to make understandable how dual imaging may improve diagnosis.

Response: We have reorganized the Results section to enhance clarity and emphasize the key findings of our study. The revised presentation now clearly distinguishes between the results for PD and atypical parkinsonisms and provides a more detailed explanation of how dual FDOPA-FDG PET/CT contributed to improving the diagnostic process. Tables and figures have also been updated to reflect these changes more effectively.

  1. A paragraph on the limitations is lacking; for example, the choice of “blinding” the nuclear medicine physicians may add, in some ways, value to the data, but this blinding does not happen in clinical practice and this may introduce potential biases (since the images were “qualitatively” evaluated, some features may have been over or underestimated, since the nuclear medicine physician did not know what to focus on).

Response: We have added a paragraph to the Discussion section addressing the limitations of our study. This new section includes a discussion of the potential biases introduced by the blinding of the nuclear medicine physicians, particularly concerning the qualitative evaluation of the images. We acknowledge that while blinding can reduce certain types of bias, it may also lead to the under- or overestimation of imaging features in a clinical setting where such blinding does not typically occur.

  1. There is no clear indication on which “current clinical criteria of each parkinsonian syndrome” were used (page 2, lines 71-72); please cite the clinical criteria in the methods or discuss in the limitations of the study the lack of formal evaluation of the diagnosis accordingly to clinical criteria.

Response: We have included specific references to the clinical criteria used for each parkinsonian syndrome in the Methods section.

  1. I also suggest the Authors to collaborate with a neurologist in order to improve the manuscript, since some clinical data (e.g. disease duration) that may be important to interpret the results are lacking and the results need to be interpreted by a person with clinical expertise. A multidisciplinary team is fundamental not only in clinical practice but also for research purposes if you want to add clinical utility to your data.

Response: We fully agree with your suggestion regarding the involvement of a neurologist. We have collaborated with a neurologist to refine the clinical interpretation of our data and to incorporate additional relevant clinical information, such as disease duration, into our analysis. This collaboration has strengthened the manuscript by ensuring that the results are interpreted with appropriate clinical expertise.

Other comments:

  1. a) We have revised the terminology throughout the manuscript to consistently use "PD" and "atypical parkinsonism" instead of "Typical Parkinson’s." The term "Parkinson’s type" has been removed from Table 1, which now only distinguishes between "PD" and "atypical parkinsonism."
  2. b) The figures have been updated to improve clarity. We have replaced the previous graphs with simpler bar graphs that use a black, white, and grey color scheme to avoid visual confusion. Additionally, the graphs have been enlarged and rendered with higher resolution to ensure they are clear and easily interpretable.
  3. c) The ellipsis ("...") in the introduction has been removed, and the sentence has been revised for clarity.
  4. d) We have corrected the references to ensure that the numbering in the text corresponds accurately to the reference list. All citations have been reviewed and corrected accordingly.

We believe that these revisions have significantly improved the manuscript and addressed the concerns you raised. We are grateful for your constructive feedback and are confident that the revised version of the manuscript is now more robust and ready for consideration for publication.

Thank you again for your thorough review.

Sincerely,

Fabio Andres Sinisterra

Reviewer 3 Report

Comments and Suggestions for Authors

In this paper the authors highlight a specific and sometimes troublesome aspect in the diagnosis of atypical parkinsonism. Despite bringing some practical concepts that can be easily applicated in the clinical practice, I am a little bit doubtful about the novelty of the paper (it is already known that sometimes a discrepancy may exist between clinical and radiological evaluations in terms of diagnosis). Maybe authors should stress more the possibile novelty and/or scientific contribution of this paper. 

Nevertheless, I have other conserns: 

- English language must be revised: sometimes articles are missing, syntax is not always correct (i.e. commas placed in wrong places).

- Table 3 is not easy to understand. Can you make it easier? Moreover a non specified abbreviation is reported (DCL: what does it stand for?)

- in some cases diagnosis reported under the figures are not completely correct. Where they based on a single physicians or was there an intern consent/agreement among different clinicals about the possible clinical diagnosis? This would be importat since you should be "as sure as possible" about clinical diagnosis in order to discuss about concordance between clinical and radiological findings. 

Comments on the Quality of English Language

As stated above, English language must be revised. Sometimes articles are missing and some sentences may be not easy to understand. Moreover, I would suggest a revision of the syntax (i.e. comma placed in the wrong places or unsell commas used)

Author Response

Dear Reviewer,

We sincerely appreciate your detailed and valuable comments on our manuscript. Below, we address each of your points and describe the changes made to enhance the quality of our work.

  1. Novelty and Scientific Contribution:

Comment: The reviewer expressed doubts about the novelty of the paper, as it is already known that discrepancies may exist between clinical and radiological evaluations in terms of diagnosis. The reviewer suggested emphasizing more on the possible novelty and/or scientific contribution of this paper.

Response: We acknowledge your concern regarding the novelty of our work. While it is recognized that discrepancies between clinical and radiological evaluations can occur, our study aims to provide a structured analysis of how dual tracer PET/CT imaging with FDOPA and FDG can enhance diagnostic accuracy in the differential diagnosis of parkinsonian syndromes. These tracers have generally been studied individually, but each has diagnostic limitations when addressing parkinsonisms. For example, FDOPA alone cannot distinguish between Parkinson's Disease and atypical parkinsonisms, while FDG can differentiate between these conditions only after confirming presynaptic dopaminergic pathway involvement. To further emphasize the novelty, we will expand the discussion section to highlight how our findings contribute to refining clinical decision-making and improving diagnostic algorithms by integrating dual-tracer imaging into routine clinical practice.

  1. English Language Review:

Comment: The reviewer indicated that the English language needs revision, as there are missing articles, syntax issues, and misplacement of commas.

Response: We appreciate your concern regarding the quality of the English language. We will carefully review the manuscript, with particular attention to missing articles, syntax, and punctuation. This will ensure that the language is clear and precise.

  1. Clarity of Table 3:

Comment: The reviewer found Table 3 difficult to understand and mentioned an unspecified abbreviation (DCL).

Response: We understand the reviewer's difficulty with Table 3, and we will revise it to present the data more intuitively. We will clarify the abbreviation "DCL" and ensure that all abbreviations are fully defined in the table legend. The table will be redesigned to improve its readability and comprehension.

  1. Accuracy of Diagnoses Under Figures:

Comment: The reviewer pointed out that in some cases, the diagnosis reported under the figures may not be entirely accurate. The reviewer also inquired whether these diagnoses were based on a single physician's opinion or if there was a consensus among multiple clinicians.

Response: Thank you for highlighting this potential issue with the diagnoses reported under the figures. The clinical diagnoses were established through consensus by a panel of neurologists specializing in movement disorders, ensuring the highest possible accuracy. We will review the figure legends to clarify this point and ensure that all reported diagnoses reflect this consensus accurately.

Additional Comments on the Quality of the English Language:

Comment: The reviewer suggested revising the English language, particularly regarding the placement of commas and the use of articles.

Response: We have noted your suggestions for improving the quality of English in the manuscript. We will make the necessary revisions to enhance the clarity and professionalism of the language used.

Thank you again for your thorough review.

Sincerely,

Fabio Andres Sinisterra Solís.

Round 2

Reviewer 1 Report

Comments and Suggestions for Authors

Authors addressed most of my comments, however in lines 68-73 in the context of tau radiotracers authors could provide a concise statement regarding their off-binding properties e.g. MAO binding

Ref.

a) Accumulation of Tau Protein, Metabolism and Perfusion-Application and Efficacy of Positron Emission Tomography (PET) and Single Photon Emission Computed Tomography (SPECT) Imaging in the Examination of Progressive Supranuclear Palsy (PSP) and Corticobasal Syndrome (CBS). Front Neurol. 2019 Feb 14;10:101. doi: 10.3389/fneur.2019.00101. PMID: 30837933; PMCID: PMC6383629.

b) The tau positron-emission tomography tracer AV-1451 binds with similar affinities to tau fibrils and monoamine oxidases. Mov Disord. 2018 Feb;33(2):273-281. doi: 10.1002/mds.27271. Epub 2017 Dec 26. PMID: 29278274.

Author Response

Coment 1: Authors addressed most of my comments, however in lines 68-73 in the context of tau radiotracers authors could provide a concise statement regarding their off-binding properties e.g. MAO binding.

Response 1: We appreciate your suggestion and, kindly following your comment, we will add a concise statement in the indicated lines to address the off-binding properties of tau radiotracers. This addition highlights how some tracers, such as AV-1451, bind both to tau fibrils and to MAO, which may affect the specificity of the images. We hope this inclusion satisfactorily addresses your comment.

Reviewer 3 Report

Comments and Suggestions for Authors

The authors well adressed all my comments

Author Response

Thank you for your positive feedback. We are pleased that you found our manuscript satisfactory and that all previous comments were addressed. We appreciate your time and effort in reviewing our work.